# Estimating the Benefits of Korea's Intercity Rail Speed Increase Project: An Agent-Based Model Approach

Chansung Kim [1,*], Heesub Rim [2], DongIk Oh [3] and Dongwoon Kang [1]

[1] SOC Digital Center, Korea Transport Institute, 370 Sicheong-daero, Sejong City 30147, Korea; dwkang7@koti.re.kr

[2] Department of Civil, Environmental, and Construction Engineering, University of Central Florida, 12800 Pegasus Drive, Orlando, FL 32816, USA; hrim@knights.ucf.edu

[3] T-Lab Consultant, Sadang Dong, Seoul 07019, Korea; cyber7511@naver.com

[*] Correspondence: cskim@koti.re.kr; Tel.: +82-44-211-3127; Fax: +82-44-211-3224

**Abstract:** In the cost–benefit analysis of urban transportation investment, a logsum-based benefit calculation is widely used. However, it is rarely applied to inter-regional transportation. In this study, we applied a logsum-based approach to the calculation of benefits for high-speed projects for inter-regional railways in Korea's long-term transportation plan. Moreover, we applied a behavioral model in which an agent travels beyond the zones assumed by an aggregate model. In the case of South Korea, such a model is important for determining transportation priorities: whether to specialize in mobility improvement by investing in a high-speed railway project, such as the 300 km/h Korea Train eXpress (KTX), or to improve existing facilities, such as by building a relatively slower railroad (150–250 km/h) to enhance existing mobility and accessibility. In this context, if a new, relatively slow railroad were constructed adjacent to a high-speed railroad, the benefits would be negligible since the reduction in travel time would not sufficiently reflect accessibility improvements. Therefore, this study proposes the use of aggregate and agent-based models to evaluate projects to improve intercity railway service and conduct a case study with the proposed new methodology. A logsum was selected to account for the benefits of passenger cars on semi-high-speed and high-speed railroads simultaneously since it has been widely used to estimate the benefits of new modes or relatively slow modes. To calculate the logsum, this study used input data from both the aggregate and individual agent-based models, and found that an analysis of the feasibility of inter-regional railroad investment was possible. Moreover, the agent-based model can also be applied to inter-regional analysis. The proposed methods are expected to enable a more comprehensive evaluation of the transport system. In the case of the agent-based model, it is suggested that further studies undertake more detailed scenario analysis and travel time estimation.

**Keywords:** aggregate; individual; logsum; agent-based model; benefit

## 1. Introduction

While the development of the previous transportation system has focused on improving mobility, recent research into new transportation services that can move people closer to their destination is in progress. In other words, methods for improving accessibility, the ultimate goal of a transportation system, are being studied. In this context, several specialized methods to improve accessibility have been introduced, such as autonomous cars, the slow transportation mode (Kim et al. [1], Standen et al. [2]), and autonomous minibuses (Kim et al. [3]), which play an important first-mile and last-mile role in urban transportation. In a wider sense, these considerations for moving closer to the final destination are required for intercity transportation as well as interregional train travel. While the operation of high-speed trains, such as the Korea Train eXpress (KTX) in South Korea, can dramatically reduce the travel time between regions, they may have lower accessibility to

the final destination compared to that of moderate-speed trains. Furthermore, since the construction and operating costs of high-speed trains are comparatively higher, users need to pay more to use them. Therefore, by improving existing moderate-speed trains, it is possible not only to reduce travel time, but also provide a low-cost service so that more people can travel on intercity trains.

In South Korea, the construction of high-speed railways has improved the mobility of intercity traffic. Recently, however, the issue of whether to construct a new high-speed railway or improve mobility by upgrading existing moderate-speed railways (MOLIT [4]) has entered the country's transportation master plan. Of the two objectives, reduced travel time has traditionally been considered the main benefit. Although many countries have applied a similar method, the problem of applying it was identified in Germany (Nagel et al. [5], Winkler [6]), where a long-term transportation plan was established using a consumer surplus-based benefit estimation method rather than a reduction in travel time. Furthermore, this method has already been used in the U.S., Netherlands, and U.K. (Ma et al. [7], Villanueva et al. [8], Geurs et al. [9], Geurs et al. [10], de Jong et al. [11], and UK DOT [12]).

In addition to the issue of which method to use to estimate the benefit of a transportation system, there is also the controversial issue of the data used in the benefit estimation. In general, aggregated data, such as the average travel time between the origin and destination, are used, but the reliability of travel times has recently come into question; moreover, few studies have been conducted into estimating benefits using disaggregated data based on big data (Kuhnel et al. [13]). Most of these mainly focus on urban transportation, while only a few focus on inter-regional transportation (Bischoff et al. [14]).

This study proposed a methodology for evaluating the impact of an improved inter-regional railroad in Korea based on two contextual points: the benefit estimation method and data types. The model developed for the benefit estimation uses the logsum estimation method, which can consider the utility of both high-speed and moderate-speed rail, including passenger vehicles. It has been already used in cost–benefit comparisons for urban transportation planning, but has not yet been applied to inter-regional transportation. Furthermore, this study presented the result of estimated benefits using two different types of input data, aggregated and individual travel times.

To sum up, the major contributions of this paper can be categorized into two parts. The first is the rarely employed logsum-based benefit estimation model for inter-regional transportation; the second is an agent-based travel demand. Paradigms have been shifting from aggregate to individual data, such as big-data-based traffic demand models that use Call Detail Record (CDR) data. Recent travel-demand models have attempted to address individual travel behavior by using an agent-based model, but results are hard to obtain in a traffic analysis zone (TAZ)-based model.

In the benefit analysis of transportation investment for high-speed inter-regional railways in Korea's long-term transportation plan, logsum-based benefit calculation techniques were successfully applied. Moreover, this study also applied a behavioral model for an individual agent. In conclusion, the agent-based model provided more specific behavioral conditions that can be accounted for during the benefit estimation. In other words, it can be extended to an analysis over specific time periods, such as the pandemic, restrictions on the use of specific links, and responses to railway accidents in certain locations [15,16]. These are hard to obtain by using the aggregate model alone.

This paper is composed as follows. In Section 2, the authors provide literature reviews focusing on the logsum-based benefit estimation and case studies. Section 3 introduces the data, analysis methodology, and scenarios. The results and discussions are explained in Section 4. Finally, in Section 5, the conclusion and suggestions for further study are described.

## 2. Previous Reviews

A public transportation investment project needs to go through a cost–benefit analysis (CBA). During evaluation, most countries adopt reduction of travel time as the great-

est benefit. For instance, suppose that the government invests in a railroad to alleviate congestion in a road network where the traffic volume between a certain origin and destination is 1000 people per hour, and travel time when using a passenger car is 30 min. After the construction of a new railroad, even though the train is slower than the car (e.g., 40 min), assume that 200 users change from car to rail. The traffic volume of the roadway decreases to 800 people per hour, and the travel time is also reduced (e.g., to 28 min). In this situation, the total travel time after the construction of new railroad is 30,400 min (800 people × 28 min + 200 people × 40 min), which is greater than doing nothing (1000 people × 30 min = 30,000 min). Therefore, although, the road congestion is reduced, this project might have negative benefits.

Similar methods for calculating benefits have been used for a long time in Korea and other countries. As described in the previous example, there are several limitations to the expected benefits. One limitation is the tendency to underestimate the benefits of public transportation investment. To address this, a methodology using consumer surplus was developed. The rule-of-half method was used to simplify calculations, but it is limited in its ability to consider new modes and induced demand. Therefore, new methods of evaluating the economics of transportation investment by using logsum as a benefit measure are being applied in the Netherlands and Germany, and the Metropolitan Planning Organization (MPO) in the United States (Geurs et al. [9], Geurs et al. [10], de Jong et al. [11], Geurs et al. [17], Nagel et al. [5], and Villanueva et al. [8]).

For example, Geurs et al. [17] proposed the mode–destination choice model for the complex analysis of land development and transportation development. The advantage of this model is that by considering both processes at the same time, it excludes the feedback process between the mode and destination choice. They suggested that the result of the logsum estimation indicated that accessibility, travel demand, and travel effect can be analyzed either separately or simultaneously. In the US MPO (Villanueva et al. [8]), this method has been more commonly applied to various investment policies, such as the expansion of roadways, construction of new roadways, decrease in transit interval, and new public transportation routes. In Germany (Nagel et al. [5]), reduced travel time has been used to evaluate a long-term transportation plan similar to South Korea's, but the consumer surplus estimation method based on economic theory was used in the recent development plan. Handy and Niemeier [18] and Niemeier [19] developed the mode–destination choice model and analyzed travel behavior in Seattle, Washington. They compared the consumer surplus before and after the removal of specific transportation modes or areas. The difference indicated the net benefit.

In addition to the development of models for estimating benefits, studies of the influence of travel time uncertainty have been conducted with different data types based on an aggregation level. In general, the average travel time between zones is used as the input data of an urban transportation or inter-regional transportation planning model. However, since traditional models use aggregated data, such as average travel time, their limitation is that they cannot analyze the travel time uncertainty, especially for inter-regional transportation because it covers longer distances than does urban transportation. Therefore, inter-regional traffic has a higher uncertainty probability. Some studies have been conducted to address this problem by combining cell phone data to the origin–destination matrix. In addition, some studies used agent-based modeling to compare the difference between using aggregated travel time and disaggregated travel time (Kuhnel et al. [13]). This study used individual travel time data collected from an agent-based simulation program instead of an average travel time.

## 3. Data and Methodology

### 3.1. Study Area

This study applied the proposed methodology over all of South Korea, which has a population of about 50 million. It has seven metropolitan cities and eight provinces and can be subdivided into 250 Traffic Analysis Zones (TAZs). The government of South Korea is

considering a long-term investment plan to increase from 50 to 80% the number of people able to travel over all the regions within three hours by 2040 (MOLIT, 2021). According to this plan, the current highway will be expanded from 4848 to 6750 km, and the railway from 5366 to 7599 km. For the railway, the construction of semi-high-speed rail is a significant portion of the investment. Current and future intercity high-speed roadway and railway network of Korea are shown in the below Figure 1.

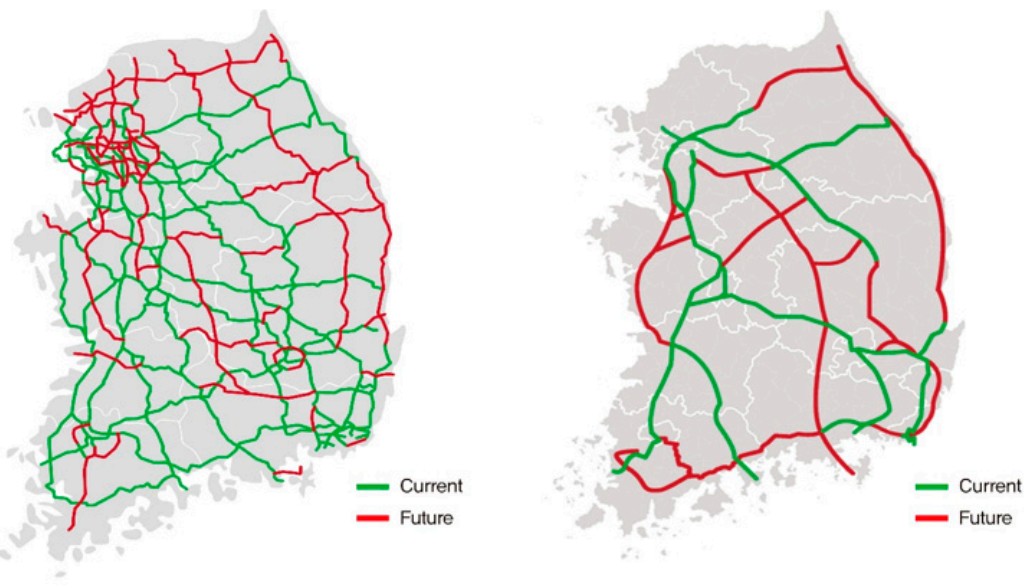

**Figure 1.** Current and future intercity high-speed roadway and railway network.

Although the effect on travel time reduction from railway investment is promising, the cost–benefit feasibility is somewhat questionable. Therefore, the proposed method should be reviewed using economics-based consumer surplus.

*3.2. Data*

3.2.1. Basic Input Data and Data Conversion Process

For input data, this study used the Korea Transport Database (KTDB) (KOTI [20]), which contains traffic volume data between TAZs subdivided into around 250 sections by mode: passenger car, bus, moderate-speed railroad, and high-speed railroad. In addition, KTDB provides not only traffic volume data but also transportation network analysis and parameters for volume delay function (VDF); thus, travel time for each origin–destination pair can be estimated. In this study, the base year for estimating the benefits of transportation projects is 2016, and the future year is 2040. The total number of trips in the base year is 85 million trips/day.

Estimating railway travel time requires information on stations, routes, operating patterns, timetables, and road networks for access to railroad links in the network data, which contains information on the length of the link and type of railway: subway, metropolitan railway, general railway, or high-speed railway. Each link in the road network data contains information such as link length, number of lanes, lane capacity, and toll road weight. Railway operation data provide information such as vehicle type, the number of operations, average operation speed, stopped/unstopped station, arrival and departure times, and operation interval. Finally, train operation data provide information for each stop pattern even on the same route.

This study used two different types of travel time estimations. First, this study adopted EMME4 (INRO [21]), a multimodal transport planning software program traditionally used in traffic demand analysis to estimate macroscopic-level travel time, which is aggregated

average travel time. The other method was to estimate agent-level travel time using MATSim (Horni et al. [22]), an open-source framework for implementing agent-based transport simulations. The network and transit schedule data provided by KTDB were in the form of EMME inputs, so data conversion to EMME format was not necessary. To estimate travel time using MATSim, the data provided by the KTDB had to be converted into MATSim input data. Basically, it consisted of network data, transit schedule data, and an individual person's trip plan.

3.2.2. Rail Travel Time Generation

This study defined travel time as the total time from origin to final destination. If someone is mainly traveling by rail, total travel time can be divided into four categories, (1) access time from the departure point to the railroad station, (2) waiting time at the railroad, (3) in-vehicle travel time, and (4) access time from the railroad station to the final destination.

To estimate each category, this study proceeded as follows. First, the in-vehicle travel time may have different values, even for the same route, depending on the performance of the train and the number of stops. Fortunately, as mentioned above, the train operation data provides detailed information such as train type and train stopping pattern. In addition, an important factor in the in-vehicle travel time is the route choice. Although there are various path selection methods, in this analysis an optimal strategy model was applied to the aggregate-level analysis, and a co-evolutionary algorithm to the individual-level analysis.

Second, the waiting time at the railroad is related to the train interval. The KTDB train operation data contains information on the number of train operations, and the average interval of the train, which in this study was applied to waiting time.

Finally, the approach time from the departure point to the railway station and the access time from the railway station to the destination can have a great influence on the approach time depending on the method used. Moreover, depending on whether the station is located in a downtown or suburban area, the difference in access time may be large. However, since the KTDB provides information only on the main mode, this study assumed an access time of 30 min for the aggregate analysis (EMME). In the individual analysis (MATSim), the time the agent took to approach the railway station was individually analyzed and applied.

*3.3. Methodology*

3.3.1. Model and Estimation Process

Residence and workplace relocations were generally described in the literature as spatial interactions. Numerous studies on spatial interactions in human behavior have examined shopping and work trip flow, human migration, residence location, firm location, recreational sites, and freight flow. In the transportation planning field, destination choice models were used when analyzing each traveler's destination.

McFadden [23] employed a well-known disaggregate multinomial logit model. Unlike most discrete choice studies, which are based on limited alternative choice sets, location choice studies include alternative choice sets. McFadden [24] suggested that in residence location studies the simplified multinomial logit model could be estimated using a randomly selected choice set. McFadden [24] and Daly [25] introduced aggregate destination choice modelling by incorporating size variables (e.g., employment, area, population, and accessibility by TAZ) as destination attributes.

The first step to calculating the logsum is to establish the mode–destination choice model. In this study, the model was derived using origin–destination traffic data by mode. To build the input data for model development, 4000 trips were randomly sampled from total trip data (85 million trips/day). The selected data consist of origin, destination, and mode. From these datasets, 10 alternatives (destination–mode pairs) were randomly selected based on each corresponding origin zone. Some studies suggest that choice sets

with random selected elements can provide reasonable results in situations with many alternative choices (Niemeier [19] and McFadden [24]).

The next step was to calculate travel time and cost by modes, and the attraction for each alternative including the information whether selected or not. Although this study did not consider all possible alternatives, according to the recent studies the model is not biased because the alternatives were randomly sampled. Due to the simplicity of the choice set, the alternative specific constant term and alternative specific variables could not be included in the formula. Only generic variables such as travel time and size variables could be accommodated. Here, assuming a Gumbel distribution of the error term ($\epsilon$), the utility ($U$) of an individual choosing TAZ $j$ by an individual in TAZ $i$ may be expressed as:

$$U_{ij} = V_{ij} + \epsilon_{ij} \tag{1}$$

$$V_{ij} = \alpha X_{ij} + \beta S_j \tag{2}$$

where $\alpha$ and $\beta$ are estimated parameters; $X_{ij}$ are the generic variables (travel time and cost) between TAZ $i$ and TAZ $j$; and $S_j$ are the size variables (total destination trips) by TAZ $j$.

### 3.3.2. Logsum Benefit Estimation

After the development of the two models, aggregated and non-aggregated, the consumer surplus (expected benefit of the proposed project) can be estimated. The estimation of mode–destination choice models are shown in the below Sections 4.1 and 4.2. To calculate the benefits, origin–destination (O–D) trip tables of the base and future years are required, for which this study used the KTDB. In addition, travel time and cost by modes between regions in the base and future year were estimated. By applying Equations (1) and (2) above and Equations (3) and (4) below at the same time, the logsum-based benefits were calculated.

The logsum-based method can simultaneously account for the benefits of passenger car, semi-high-speed rail, and high-speed rail since it has been widely used to estimate the benefits of new modes or relatively slow modes.

To estimate consumer surplus, logsum ($L_i$) and travel demand ($O_i$) for both base and future years were calculated. After that, the logsum difference between the base and the future year, considering travel demand, was estimated. Finally, the consumer surplus was estimated by multiplying the logsum difference by the value of travel time. The estimation of consumer surplus is shown in Equations (3) and (4). The subscripts "before" and "after" indicate the base and future year, respectively. This study assumed that the travel demands of both years were identical. The following Section 4 introduces the results in detail.

$$L_i = \log(\sum_j \exp(V_{ij})) \tag{3}$$

$$CS = VOT \times \{[O_i \times \log(\sum_j \exp(V_{ij})]_{After} - [O_i \times \log(\sum_j \exp(V_{ij})]_{Before}\} \tag{4}$$

### 3.3.3. MATSim Application for Benefit Estimation

A simulation methodology based on the autonomous car and minibus was applied, and it evolved a simulation which was used to estimate changes resulting from the introduction of an autonomous vehicle-based transportation system. An unmanned, autonomous taxi was selected as the target service, and MATSim was selected as the simulation tool. MATSim was developed as open source implemented in Java, so it is easier to use for various research purposes than other simulation tools.

The MATSim simulation process consists of five modules: initial demand (Module 1), simulation (Module 2), scoring (Module 3), analysis (Module 4), and replanning (Module 5), which are optimized through repeated computations [22].

MATSim uses a co-evolutionary algorithm to optimize the plan for each agent's activity and trip. This algorithm executes the agent's plan and searches for alternatives that can be improved according to the constraints. It then maintains a stable state when it can no

longer be improved. The algorithm is also composed of a queue-based traffic simulation to increase efficiency for large-scale scenario analysis.

An agent's travel time for model estimation and average travel time for benefit calculation were extracted using MATSim. For model estimation, this study used the travel time and cost between origin and destination by modes for each individual. Further, the average value of travel time and cost by mode between origin and destination was used for benefit estimation. To extract travel time, the event file generated by MATSim was used. The travel time of all agents was extracted from the event file, using Python. To reduce the burden of excessive calculation, this study randomly sampled 4000 people, and the number of simulation iterations was limited to 50.

## 4. Results and Discussions

### 4.1. Estimating the Benefit of Intercity Rail Speed Increase: Conventional Method

The result of the mode–destination choice model is presented in Table 1. Considering the travel diary survey, the number of samples this study extracted (4000) was sufficient to estimate a significant model (NLOGIT6 [26]). The variables used in the model estimation were travel time and cost, whether or not the person arrived in Seoul (CBD), and attraction (total destination trips). This study used NLOGIT6 to estimate the mode–destination choice model, and the result is presented in Table 1. Considering maximum likelihood estimates (MLEs) between before and after convergence, the estimated model was statistically significant. In addition, as shown in Table 1, the model was logically significant for the sign of the coefficient. As travel time and cost decreased, utility increased, and the coefficient of attraction was positive. Therefore, the estimated model is significant.

**Table 1.** Estimated parameters of mode and destination choice model (conventional).

|          | Coeff.    | Std. Err. | t-Ratio | *p*-Value |
|----------|-----------|-----------|---------|-----------|
| TIME     | −0.02442  | 0.00094   | −26.08  | 0.000     |
| COST     | −0.00003  | 0.00001   | −4.03   | 0.000     |
| CBD      | −0.28736  | 0.06031   | −4.76   | 0.000     |
| ATTRACT  | 0.00001   | 0.00000   | 10.31   | 0.000     |

Note: Log likelihood value at convergence is −8109.9, Log likelihood value at zero is −9210.3, and $\rho^2 = 0.1195$.

Using the estimated mode–destination choice model, the logsum of the base year and future year were calculated, and the consumer surplus was calculated by applying the value of time and is presented in Table 2. The benefit can be calculated by comparing consumer surplus between the base and the future years. This study assumed that the total traffic volume in the base year and the future year would be the same. As shown in Table 2, the annual benefit from Korea's intercity rail speed increase is estimated at 76 billion USD. Therefore, the proposed method in this study, estimating the benefit by using logsum, is applicable to the evaluation of Korea's intercity railroad. As shown in Table 2, the mixed consumer surplus indicates the benefit from simultaneous road and railroad improvements. The net consumer surplus indicates the benefit from when only either the road or railroad was improved.

**Table 2.** The benefits of intercity rail speed increase project (Conventional).

|                      | Benefits (USD) |
|----------------------|----------------|
| Mixed Benefits       | 111 billion    |
| Net Benefits (rail)  | 76 billion     |

### 4.2. Estimating the Benefit of Intercity Rail Speed Increase: Abm Method

Table 3 shows that the disaggregated model estimation result using individual travel time selected from MATSim. Considering the large difference between MLEs before and after convergence, the estimated model was statistically significant. Furthermore, the sign

of the coefficients was also reasonable. The coefficient of time and cost had negative values and attraction was positive. These results were consistent with those in Section 4.1.

**Table 3.** Estimated parameters of mode and destination choice model (ABM).

|         | Coeff.   | Std. Err. | t-Ratio | *p*-Value |
|---------|----------|-----------|---------|-----------|
| TIME    | −0.01027 | 0.00057   | −18.04  | 0.000     |
| COST    | −0.00010 | 0.00001   | −13.42  | 0.000     |
| CBD     | −0.37724 | 0.05761   | −6.55   | 0.000     |
| ATTRACT | 0.00001  | 0.00000   | 8.97    | 0.000     |

Note: Log likelihood value at convergence is −8319.3, Log likelihood value at zero is −9210.3, and $\rho^2 = 0.09674$.

Using the model presented in Table 4, the consumer surplus (benefit) from Korea's intercity rail speed increase project was estimated at 73 billion USD. Therefore, this result shows that the method based on disaggregated travel time was also applicable to the evaluation of inter-regional transportation. However, it was found that the value of the parameter affecting the benefit and the amount of the benefit was different.

**Table 4.** The benefits of intercity rail speed increase project (ABM).

|                    | Benefits (USD) |
|--------------------|----------------|
| Mixed Benefits     | 94 billion     |
| Net Benefits (rail)| 73 billion     |

*4.3. Comparison between Aggregated and Disaggregated Models*

4.3.1. Comparison of Travel Time

Before the analysis of consumer surplus, this study investigated travel times by different methods of estimation. This study estimated travel time using three methods: EMME 4, MATsim, and open API. The results showed that the distributions of average travel time were comparable. However, although the origin and destination were identical, the estimated travel time using MATsim and open API (Daum [27]) were slightly different as shown in Figure 2. In this figure, the horizontal and vertical axes indicate the travel distance and the travel time, respectively.

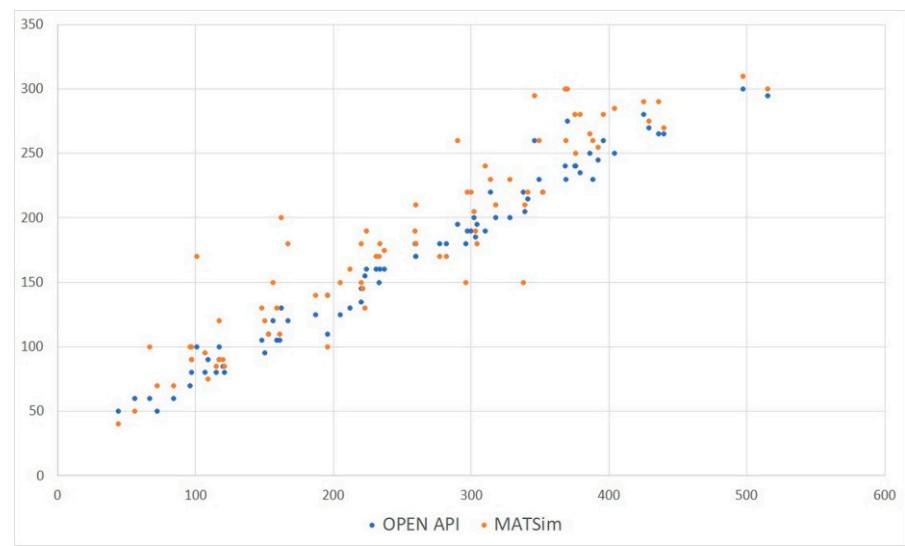

**Figure 2.** Scatter gram for travel time (minutes) for selected origin–destination pairs.

4.3.2. Results of the Estimated Parameters and Benefits

Tables 1 and 3 show that the goodness of fit of the aggregated model outperformed that of the disaggregated model in this case study. The reason was that this study used a very

large-scale intercity transportation model, so it is suspected that the MATSim limitation of 50 iterations to reduce the run time affected the predictive power of travel time. Further studies are needed on this limitation.

To estimate the benefit of the railroad improvement plan, traditional multi-modal average travel time and multi-modal average travel time from individual data were employed for an aggregated model (using EMME 4) and a disaggregated model (using MATSim), respectively. The results showed that the estimated benefits from each model were slightly different. The difference was not significant in the case of the rail project, but was significant for the total benefits of the road and rail project. Therefore, it is noteworthy that travel time was the significant input data of benefit estimation. Furthermore, the results showed the importance of accurate estimated travel time, and it is emphasized that research into the development of a multi-modal travel time estimation method is needed.

## 5. Discussions

In the cost–benefit analysis of urban transportation investment, the logsum-based benefit calculation has been widely used. However, it is rarely applied to inter-regional transportation. This study applied it to the calculation of benefits for high-speed projects for inter-regional railways in Korea's long-term transportation plan. Moreover, this study attempted to apply a behavioral model in which an individual agent traveled beyond the zonal assumptions of an aggregate model. It was a successful experiment in some respects. First, the results showed that it was possible to analyze the feasibility of inter-regional railroad investment with both aggregate and agent-based models. This indicated that the agent-based model can also sufficiently account for inter-regional analysis and be applied to it.

However, some limitations were found. In the case of an agent-based model, the number of iterations had to be limited to 50 to reduce the computational burden; thus, a travel time value with insufficient stabilization might be used. While the net benefits of railroad investment were similarly calculated, the mixed benefits of road and railroad investment differed greatly, and it is necessary to clarify the difference. Therefore, in the case of the agent-based model, it was suggested that more detailed scenario analysis, travel time estimation, and further studies are recommended on route and path analysis by mode.

From the application perspective, the agent-based models enabled the analysis of positive or negative benefits for more behavioral situations. In other words, it can be extended to research in areas such as specific time periods, restrictions on the use of specific links, and responses to railway accidents in certain locations. Since travel behavior can be affected by these events, the proposed methodology, which considers individual behavior, would be suitable to assess these effects.

## 6. Conclusions

Many countries are improving mobility by enhancing existing facilities, and the benefit estimate from a traffic system improvement project has become an important issue in evaluating intracity and intercity transportation systems. This study proposes a methodology for estimating the benefit from a transportation system improvement project and evaluated Korea's railroad speed increase project using the proposed method. The model developed for the benefit estimation was derived from a change in consumer surplus using logsum, which accounts simultaneously for the benefits of passenger car, low-speed railroad, and high-speed railroad.

To analyze the effect of travel time, two types of data were used: traditional aggregated data of inter-zone travel time, and individual travel time information obtained from an agent-based model. From our analysis, it was found that the benefits in intercity and intracity transportation can be estimated by the logsum method.

Moreover, it was found that the effect on the benefits of travel time and cost and destination trip volume was considerable. The impact was found to be large for aggregated data but less for disaggregated data. The net benefit of railroad investment was similarly

calculated as 76 billion USD for the aggregated model and 73 billion USD for the agent-based model, while the mixed benefits of road and railroad investment differed greatly. Therefore, comparative studies are recommended for various scenarios, such as differences in a route. In addition, in the case of the agent-based model, although the travel time extracted after setting the number of simulations to 50 was used, it is suggested that further studies are needed to analyze the effect of the number of iterations.

In the benefit analysis of transportation investment for high-speed projects for inter-regional railways in Korea's long-term transportation plan, logsum-based benefit calculation techniques were applied as was an agent behavioral model. The results of this study are not conclusive, but challenging. In the future, we propose further study to account for the reliability of travel time when estimating the mode–destination choice model in intercity travel. In addition, it is also recommended that a model be developed to account for various travel purposes and income levels.

**Author Contributions:** The authors confirm contribution to the paper as follows: conceptualization: C.K., methodology: C.K.; software: D.O.; formal analysis: C.K., D.O.; writing—original draft preparation: C.K., H.R., D.K.; writing—review and editing: C.K., H.R.; visualization: D.K., D.O.; supervision: C.K.; project administration: C.K., D.O. All authors have read and agreed to the published version of the manuscript.

**Funding:** This research was funded by The R&D Convergence Program of the National Research Council of Science & Technology of the Republic of Korea (CAP-16-02-KIST and CRC-20-02-KIST).

**Institutional Review Board Statement:** Not applicable.

**Informed Consent Statement:** Not applicable.

**Data Availability Statement:** Not applicable.

**Conflicts of Interest:** The authors declare no conflict of interest.

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
