# Peer review of "Estimating the Benefits of Korea’s Intercity Rail Speed Increase Project: An Agent-Based Model Approach"

_2673-3951, doi:10.3390/modelling3010007_

Round 1

Reviewer 1 Report

This paper is about a simulation study of Korea's future transportation plan. The topic is interesting and the methods used are appropriate. However, there are some suggestions which can make this paper better.

Surprisingly, the abstract is not well-written. The paper otherwise is written well. 

The authors stated that: "The study utilized individual travel time data collected from an agent-based simulation program instead of average travel time. " It will be interesting for the readers if the agent-based model used is also documented so that agent types and their interaction are clearly identified.

Similarly, the numerical methods (section 3.3) are very precise. Readers would like to see details of the model used. Many readers do not have in-depth knowledge of the terms used. So, section 3.3 should be extended with more explanation.

Lastly, the quantitative outcomes/results should be mapped onto qualitative statements. In the paper here and there, some hints are given, but, a bulleted list of qualitative outcomes would be beneficial to comprehend the significance of the researc h. 

Author Response

This paper is about a simulation study of Korea's future transportation plan. The topic is interesting and the methods used are appropriate. However, there are some suggestions which can make this paper better.

Surprisingly, the abstract is not well-written. The paper otherwise is written well.

  • The abstract was revised based on the comment.

The authors stated that: "The study utilized individual travel time data collected from an agent-based simulation program instead of average travel time. " It will be interesting for the readers if the agent-based model used is also documented so that agent types and their interaction are clearly identified.

  • In the previous Section 3.3, newly 3.3.3 MATSim application section was added. And, in the revised version, the behavioral explanations were supplemented in the abstract, introduction, methodology, and conclusion sections.

Similarly, the numerical methods (section 3.3) are very precise. Readers would like to see details of the model used. Many readers do not have in-depth knowledge of the terms used. So, section 3.3 should be extended with more explanation.

  • More explanations were extended on section 3.3.1 and 3.3.2

Lastly, the quantitative outcomes/results should be mapped onto qualitative statements. In the paper here and there, some hints are given, but, a bulleted list of qualitative outcomes would be beneficial to comprehend the significance of the research.

  • The conclusion was revised based on the comment.

Reviewer 2 Report

This study adopts two different methods to evaluate the impact of the improvement of the inter-regional railroad in Korea. In addition, the differences between the two methods are compared. This study has certain significance for the benefit evaluation of the transportation system. However, I think the following suggestions are necessary for further improvement of the article:

  1. The research gap should be further clarified in the Introduction. In addition, the authors should further clarify the research aim. Why it is necessary and significant to compare the two methods?
  2. In the Introduction, it might be better to add the organization of the paper. In addition, the authors can briefly summarize your main findings.
  3. In the Methodology section, it is necessary to report the overview of the agent based model.
  4. It is better to add a discussion part before Conclusions to show the insights from results.

Overall, I think this paper is interesting; however, the authors should pay attention to my last comment.

Author Response

This study adopts two different methods to evaluate the impact of the improvement of the inter-regional railroad in Korea. In addition, the differences between the two methods are compared. This study has certain significance for the benefit evaluation of the transportation system. However, I think the following suggestions are necessary for further improvement of the article:

The research gap should be further clarified in the Introduction. In addition, the authors should further clarify the research aim. Why it is necessary and significant to compare the two methods?

  • The introduction was revised based on the comment. Some related references were added.

In the Introduction, it might be better to add the organization of the paper. In addition, the authors can briefly summarize your main findings.

  • The introduction was revised based on the comment.

In the Methodology section, it is necessary to report the overview of the agent based model.

  • More explanations were extended at section 3.3.

It is better to add a discussion part before Conclusions to show the insights from results.

  • Discussions section was added at newly section 4.4.

Overall, I think this paper is interesting; however, the authors should pay attention to my last comment.

Round 2

Reviewer 2 Report

My comments have been addressed.